# Can Exposure to Certain Urban Green Spaces Trigger Frontal Alpha Asymmetry in the Brain?—Preliminary Findings from a Passive Task EEG Study

**DOI:** 10.3390/ijerph17020394

**Published:** 2020-01-07

**Authors:** Agnieszka Olszewska-Guizzo, Angelia Sia, Anna Fogel, Roger Ho

**Affiliations:** 1Institute for Health Innovation & Technology (iHealthtech) MD6, 14 Medical Drive, #14-01, Singapore 117599, Singapore; pcmrhcm@nus.edu.sg; 2NeuroLandscape Foundation, Suwalska 8/78, 03-252 Warsaw, Poland; 3National Parks Board, Centre for Urban Greenery and Ecology, 1E Cluny Road Singapore Botanic Gardens, Singapore 259601, Singapore; ANGELIA_SIA@nparks.gov.sg; 4Singapore Institute for Clinical Sciences, Agency for Science, Technology and Research, 12 Science Drive 2, Tahir Foundation Building #12, Singapore 117549, Singapore; Anna_Fogel@sics.a-star.edu.sg; 5Department of Psychological Medicine, Yong Loo Lin School of Medicine, the National University of Singapore, NUHS Tower Block, Level 9 1E Kent Ridge Road, Singapore 119228, Singapore

**Keywords:** urban, landscape, brain, visual, green, contemplative, mental health, well-being, FAA, EEG, UGS, depression

## Abstract

A growing body of evidence from observational and experimental studies shows the associations between exposure to urban green spaces (UGSs) and mental health outcomes. Little is known about which specific features of UGS that might be the most beneficial. In addition, there is potential in utilizing objective physiological markers of mental health, such as assessing brain activity, but the subject requires further investigation. This paper presents the preliminary findings from an on-going within-subject experiment where adult participants (*n* = 22; 13 females) were passively exposed to six landscape scenes within two UGSs (a park and a neighborhood green space) and three scenes of a busy urban downtown (control site). The landscape scenes were pre-selected based on their contemplative landscape score (CLS) to represent different levels of aggregation of contemplative features within each view. Participants went to each of the sites in a random order to passively view the scenes, while their electroencephalography (EEG) signal was being recorded concurrently. Frontal alpha asymmetry (FAA) values, commonly associated with the approach-related motivation and positive emotions, were extracted. The preliminary results show trends for the main effect of site on FAA, suggestive of stronger FAA in park compared to the control site, akin to more positive mood. There was also a trend for the interaction between the site and scene, which suggests that even within the individual sites, there is variability depending on the specific scene. Adjusting for environmental covariate strengthened these effects, these interim findings are promising in supporting the study hypothesis and suggest that exposure to urban green spaces may be linked to mental health outcomes.

## 1. Introduction

Environmental exposures are a sum of all sensory stimuli we receive along the life cycle, closely intertwined with our mental health and well-being (MH&WB) [1,2]. As visual stimuli provide the most information about the environment around us [3], the visual quality of scenes we are exposed to seems worth investigating. This is even more so if we consider a rapidly urbanizing world where most people already live in cities [4].

Urbanized areas are characterized by the concentration of built elements and infrastructure, crowdedness, noise and pollution. Visually, the urbanized environment is highly transformed from its natural manifestation. Potential negative effects of the urban visual exposures on MH&WB have been already noted by Sir Fredrick Law Olmsted, the front runner of American sanitary reform in early XIX century, known as the Father of Landscape Architecture: “A man’s eyes cannot be as much occupied as they are in large cities by artificial things…without harmful effect, first on his mental and nervous system and ultimately on his entire constitutional organization” [5]. Recent meta-analysis seems to confirm these words with scientific evidence, demonstrating the 38% higher prevalence rate of mental health disorders in urban when compared to rural areas, with mood disorders, such as major depressive disorder (MDD) causing a major burden [6,7].

Therefore, urban green spaces (UGSs) play a major role in mitigating the negative effects of urban environment exposures on MH&WB. There is an established consensus among researchers that contact with natural environments has beneficial influence on MH&WB of people [8,9,10,11,12,13,14,15,16]. However, most studies are based on a vague comparison between “urban” versus “nature” exposure, while more specific elements and attributes of UGS, and their composition have not been, to date, identified. Moreover, the knowledge in this area is based mostly on the correlational analyses, and more rigorous experimental approached are needed to examine causal relationships between specific environmental features and mental health that will form the basis of future urban landscape design. There is a need to identify which specific types of the natural environments found in cities have the most beneficial effects on people’s MH&WB. Moreover, there is a need to assess these effects through rigorous experimental designs, with the use of objective tools and objective markers of mental health in order to examine causality.

### 1.1. Frontal Alpha Asymmetry and Mental Health

FAA is one of the most studied brainwave patterns that is an objective measure of current mood. It is associated with an increased alpha power in the right frontal lobe when compared to the left. As in brain science more alpha power indicates less activation, FAA can be also referred to as decreased activation of the right frontal lobe when compared to the left. According to the approach-withdrawal hypothesis [17] FAA is associated with the appetitive motivational system towards the perceived stimulus and positive emotions (e.g., happiness and calmness), while the pattern opposite to FAA is associated with aversive behavior and negative emotions (e.g., fear and sadness), and withdrawal in relation to the perceived stimulus. The latter had been observed in patients with depression [17,18,19,20,21] and comorbid anxiety [22,23] in multiple studies, and therefore recognized as a potential marker for depression [24]. The therapeutic value of inducing FAA in the brain of depressed patients, sometimes known as “alpha asymmetry training”, has been demonstrated in multiple clinical studies [25,26,27]. We argue that passive environmental exposures, depending on their features, can induce either a pattern of brain activity associated with positive emotions and approach or the aversive pattern. Being able to identify urban settings and scenes inducing the FAA, and estimate the size of their therapeutic effect, can contribute to establishing new, cost-effective, widely accessible approaches to self-care and mental healthcare promotion. Such approaches have the potential to become a new self-care practice for diagnosed patients, and also of public health benefit in preventive care for urban dwellers at risk of mental health issues or who are looking for a mental health improvement regime.

### 1.2. Scope

Previous studies on the role of UGS on MH&WB have several limitations—including the lack of reliable physiological biomarkers and tools that can be deployed for an outdoor environment. For example, a relatively inexpensive and highly portable electroencephalography (EEG) apparatus, Emotiv EPOC has been used previously but its reliability is questionable due to high noise-sensitivity and generally poor performance [28]. Furthermore, previous studies did not attempt to establish causality between the actual exposure and MH&WB outcomes, but focused on correlational relationships [29,30,31]. In this study, we leveraged on the technological advances in neuroscience by using a non-invasive, portable yet reliable brain scanning equipment. We considered the passive exposure to the views as the most common and the most accessible type of interaction with the surrounding space. Additionally, in this preliminary phase we tested the passive exposures in situ, when immersed in actual outdoor spaces to ensure the ecological validity, rather than pictorial representations in the laboratory setting. The aim of the current study was to examine the effects of exposure to UGS with varying visual quality, on the pattern of brain activity recorded using EEG neuro-imaging. Data collection is ongoing. It is anticipated that the results of this study will provide novel information on the effects of the features of urban spaces on mood and will inform future intervention and prevention programs.

## 2. Materials and Methods

### 2.1. Participants

This is an on-going study and the results presented here are based on a sub-sample of 22 healthy volunteers (13 female, age M = 32.9, SD = 12.7) from Singapore. Participants were of Chinese (73%), Indian (5%) and other (22%) ethnicities. The full sample on completion will comprise *n* = 100. The inclusion criteria were: age between 21 and 75 years old, right-handed, no serious visual impairment, no pacemaker or recent otologic surgery and willing to commit to the full study period of three outdoor sessions. Participants were reimbursed with S$50 worth vouchers after completion of the sessions. One participant did not complete the experiment, without providing a reason, and therefore was not included in the analyzed sample. The study was granted NUS Ethical Committee’s approval (ref#: S-18-352).

### 2.2. Sites and Scenes Selection

The selected sites are two UGS in Singapore (an urban park and a neighborhood green space), and one busy urban street (control) characterized by negligible greenery. Geographically, Singapore is located near the equator and has tropical climate with abundant rainfall (14 rainfall days per month), high and uniform temperatures and humidity all year round (28 °C, 70% Rh on average). Meteorological variables do not show large month to month variability and diurnal variations are strongly influenced by solar radiation [32].

Within each site, participants were exposed to three different landscape scenes with different landscape quality values and features within the view. Each of the scenes was previously annotated by four landscape architecture experts using the contemplative landscape model (CLM) and a mean was used to score each scene [33]. CLM is an expert-based instrument to assess level of aggregation of contemplative components of any given UGS within the viewing angle from the ground. There are seven key-components (and 36 possible subcomponents) of the CLM: landscape layers, landform, vegetation, color and light, compatibility, archetypal elements and character of peace and silence, each to be scored on a 1–6 Likert scale, which ranged from low to high aggregation level per component [33,34]. Higher scores were considered more contemplative. The most contemplative landscapes are considered to be soothing, restorative environments, which are inviting to rest, promote contact with nature, self-reflection and reorientation of one-self within a larger order [35,36]. Photographs of the selected scenes as well as their CL scores are presented in Figure 1.

### 2.3. Assessment of Participant’s Depression

In order to control for depressive symptoms we utilized Beck Depression Inventory II (BDI), a 21-item self-reported multiple choice inventory, widely used as an indicator of severity of depression (81% of sensitivity and 92% specificity) [37].

### 2.4. Procedure

Data was collected between March and May 2019, during morning or late afternoon hours of the working week. Experimental sessions were scheduled individually, during one session there was one site with three scenes visited (order randomized [38]). Participants were blinded, to the hypothesis. Participants completed the self-reported Profile of Mood States questionnaire (POMS [39]) before and after each experimental session. Following that, they were seated on a chair facing the selected scene and EEG apparatus, V-amp 16-channel amplifier with dry active electrodes (Brain Products GmbH, Munich, Germany), was installed on their head. The participants were first instructed to put on the white mask blocking the view and then to relax, while equipment was calibrated and raw signal recording was initiated. After 1 min recording of the resting state with the mask on, the participants were asked to remove the mask and passively observe the landscape scene before them for another 1 min. Once this was completed, the 1 min resting state with mask on and 1 min scene watching was repeated for the same scene. This process was repeated for all the three scenes. Scenes locations as well as walking areas were selected in shaded areas to avoid excessive sun exposure. After the recording for all the three scenes were over, the participant completed the post- measurement POMS questionnaire. The duration of each session took between 30 and 45 min. Participants were allowed to drink water between the scenes but not to eat. Environmental variables (temperature, humidity, brightness and noise) were recorded with 4-in-1 environment meter (CEM, DT-8820) at each scene for each participant to control for confounding variables. The procedure on the day of testing is described in Figure 2.

### 2.5. EEG Data Processing

The EEG signal was processed offline using the Brain Analyzer 2 software (Brain Products GmbH, Munich, Germany). The raw signal had a 500 Hz sampling rate, with 2000 µS sampling interval, and was filtered with the high cutoff 40 Hz and 50 Hz notch filter. Channels were referenced according to the 10–20 international system and the visual inspection was performed to evaluate the signal quality, noisy or missing channels. Topographic interpolation of noisy or lost channels was performed where necessary, and ocular movements were corrected using the Independent Component Analysis (ICA) ocular correction function. Later, the 60 s long segments for both baseline and viewing conditions were extracted and non-complex fast Fourier transformation was performed on both segments. Further, the frequencies from two segments were averaged, and the alpha power (8–13 Hz) values for each electrode was extracted. The exported viewing data was then corrected for the baseline (viewing-baseline), and alpha power values for the left and right frontal regions were computed (Left = AFp1 + AFF5h + F7; Right = AFp2 + AFF6h + F8) [17,22]. Positive FAA values are indicative of more alpha power on the right frontal hemisphere as compared to left and negative of more alpha power on the left frontal hemisphere compared to the right. Higher values indicate higher FAA and are a marker of positive approach motivation as opposed to the withdrawal.

## 3. Statistical Analysis

A repeated measures general linear model (3 × 3) was used to test the main effect of site and the interaction between the site and the view on FAA. Unadjusted post-hoc comparisons were conducted to investigate specific group differences. In addition, to examine whether the landscape score (CLS) predicts FAA independently of the experimental condition, we used a linear regression model with bootstrapped 95% confidence intervals (CI), with 1000 samples and bias corrected accelerated CIs. For this purpose, CLS from the control condition were dummy coded and assigned a score of 1, while the raw CLS from the experimental conditions remained unchanged. As external conditions may impact the quality of exposure, data were re-analyzed after adjusting for Thom’s discomfort index (TDI), which is a good summary measure of the external conditions [40]. Due to the small sample size it was not viable to control for additional environmental measures such as temperature or humidity. As the sensitivity analysis, all tests were repeated after adjusting for BDI and the results remained unchanged (not presented). Analyses were also adjusted for participant gender, and although the results did not change, there were some non-significant indications of potential gender differences in the responsiveness to the experimental manipulation (not reported here). Dependent subjects *t*-test was used to compare the before and after POMS scores. Analyses were conducted in IBM SPSS v.23 (IBM corp., Armonk, NY, USA) and the alpha level of 0.05 was used as an indicator of statistical significance.

## 4. Results

### 4.1. Sample Characteristics

Sample characteristics, including selected environmental metrics, are described in Table 1.

### 4.2. Differences between the Sites and Views

Sphericity assumption was violated for the main effect and the interaction (*p* < 0.05), so multivariate tests (Pillai’s trace) were reported alongside the within-subject effects, with degrees of freedom corrected using Greenhouse–Geisser estimates of sphericity.

The multivariate effects showed a non-significant trend for the main effect of site (F(2, 15) = 3.04, *p* = 0.078), which was attenuated in the within-subjects tests (F(1.17, 18.77) = 1.37, *p* = 0.26). After adjusting for TDI, the multivariate tests showed a significant main effect of site (F(2,12) = 5.14, *p* = 0.024), which was slightly attenuated in the within-subjects tests (F(1.15, 14.95 = 3.25, *p* = 0.087).

The highest mean FAA were recorded in Site 1 (urban park), and the lowest in Site 3 (urban street) and these are depicted in Figure 3a. Site 1 did not differ from Site 2 (neighborhood green space; *p* = 0.61) or Site 3 (*p* = 0.20), but Site 2 was significantly different from Site 3 (*p* = 0.023).

The multivariate tests showed a non-significant trend for the interaction between the site and the view (F(4, 13) = 2.57, *p* = 0.088), which was not present in the within-subjects tests (F(1.48, 23.63) = 0.36, *p* = 0.64). These trends were further attenuated after adjusting for TDI in multivariate tests (F(4,10) = 1.84, *p* = 0.20) and though within-subjects comparisons showed significant interaction between the site and the view (F(1.67, 21.69 = 5.42, *p* = 0.016). The mean FAA across the sites and the views are summarized in Figure 3b.

Independently of the site, there was a non-significant weak positive association between the CLS and FAA (β = 0.10, *p* = 0.16), and adjusting for TDI did not influence that association (β = 0.10, *p* = 0.17).

In addition, there were no POMS changes before and after the exposure in Site 1 (*p* > 0.05) and Site 2 (>0.05). However, the total mood disturbance seemed to increase from before to after the exposure in the control group (t = −2.0, *p* = 0.06).

## 5. Discussion

This study examined FAA among participants exposed to UGS varying in visual quality. The preliminary results presented here show non-significant trends suggesting that passive visual exposure to certain UGS might be linked with higher alpha power in the right frontal lobes as compared to the left, compared to exposure to other UGS views for example landscapes in the residential areas. Moreover, same participants showed reverse pattern of brain activity (more alpha on the left compared to right lobe) when exposed to the busy street environment in high-density downtown with negligible greenery. Importantly, there was a trend for a non-significant interaction between the site and the view type, suggesting that even within the individual sites, there may be a heterogeneous response to different views and some scenes may induce more or less FAA response than others (Figure 3b). This finding informed a post-hoc hypothesis that, independently of the site, views with higher CLS might invoke higher FAA. We subsequently tested this assumption and the results showed a non-significant positive trend between CLS and FAA, suggesting that even within the residential areas, views with higher CLS might be beneficial for FAA, which might provide important information when planning the landscapes of residential areas in the future. After adjusting for environmental conditions, we observed stronger effects, which suggests that not controlling for the weather conditions, especially in countries with strong temperature and humidity variability might constitute a serious confounding factor in these types of experiments.

These preliminary findings support previous research findings suggesting that quality of green exposure can play a more important role than the quantity or accessibility of UGS for the MH&WB outcomes [29,42] and build on the previous research by providing evidence for an objective improvement in brain pattern activity typically associated with positive mood/affect. Nevertheless, this is the first study to demonstrate that in-situ exposure to a contemplative space can elicit FAA. Previous experimental studies on FAA were, to our best knowledge, conducted only in the indoor laboratory environment, and demonstrated greater FAA in the brain of healthy control as compared to depressed patients [17,18,19,20,21,22,23,24]. Studies also show that besides the withdrawal tendencies associated with depression, FAA can be a marker of the positive and negative affects. For example, Davidson and colleagues [43], examined brainwaves of participants watching and evaluating television shows. Positively rated TV scenes were associated with greater relative left-hemispheric alpha frontal activation, while negatively rated scenes were associated with greater relative right-hemispheric frontal activation. Interestingly, previous within-subjects study on the brain response to contemplative landscape videos, conducted indoors, did not find FAA differences between contemplative and non-contemplative landscape exposure [44]. As for the previous experiments conducted outdoors, they did not examine the FAA patterns, but demonstrated that green urban spaces (often described as nature) triggered improved mood, emotional regulation, recovery from stress and mental fatigue as compared to exposure to space often defined as urban [45,46,47]. In our study we did not observe improved mood (as measured with POMS) between before and after the exposures in both green urban sites. However, we did observe decreased mood in the busy urban street site. This suggests that FAA may be a more accurate predictor of positive MH&WB outcomes caused by the urban landscape exposures, than the self-reported mood questionnaire such as POMS. Alternatively, it is plausible that these two measures capture different aspects of mood, which should be explored in future research. Moreover, the in-situ EEG measurement besides having more ecological validity, may also be more sensitive to differences in FAA, as compared to indoor lab exposures.

Future research will examine the differences in brain oscillations between indoor and outdoor exposure to contemplative and non-contemplative landscapes, including in the sample patients with diagnosed depression. We will also run the dose-response analysis, which will evaluate the optimal exposure time for the best MH&WB outcomes.

The strength of this study was a well-controlled experimental design with multiple conditions and objective physiological measures. The main limitation of the study was the small sample size at this preliminary stage, as the data collection was still ongoing. There also was a possibility of bias regarding the outlying FAA values, noticeable through the values of standard deviation varying between Site 1 and Site 2 and 3. However, the bias risk was mitigated by a within-subjects design, as each participant went through the same data collecting procedure in three different environments and to remain conservative all FAA values were adjusted for baseline.

## 6. Conclusions

On completion, the results of this study will provide a basis for prevention and interventions that target mental health outcomes and will be used as guidelines for designing UGS that are optimal for mental health promotion.

## Figures and Tables

**Figure 1 ijerph-17-00394-f001:**
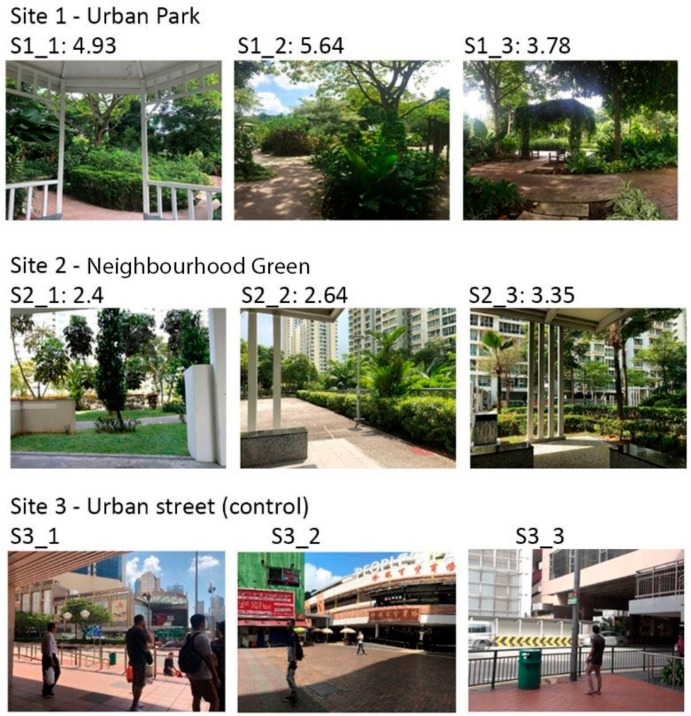
Selected sites and scenes, with contemplative landscape score between 1 and 6 points for each scene.

**Figure 2 ijerph-17-00394-f002:**
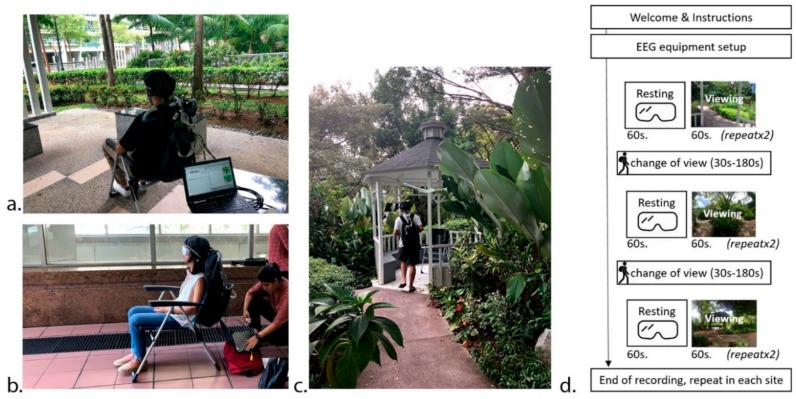
Setup and procedure: (**a**) participant viewing the scene in S2_3, (**b**) participant during resting state in S3_1, (**c**) participant walking between the scenes in Site 1 and (**d**) steps of the in-situ experimental protocol.

**Figure 3 ijerph-17-00394-f003:**
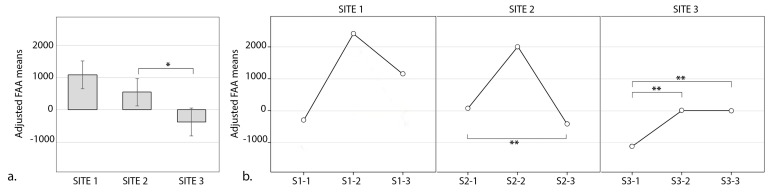
Differences in mean values of frontal alpha asymmetry (FAA) across participants passively viewing the landscapes: (**a**) within 3 sites (Site 1—Urban Park, Site 2—Residential Green and Site 3—Control (Busy Downtown) and (**b**) exposed to nine different scenes of these sites (S1_1, S1_2, S1_3, S2_1, S2_2, S2_3, S3_1, S3_2 and S3_3; * *p* < 0.05; ** *p* < 0.1).

**Table 1 ijerph-17-00394-t001:** Sample characteristics (mean, SD or #).

Variable	Participants in Preliminary Phase (*n* = 22)		
**Age**	**M = 32.9 (SD = 12.7)**		
**Gender**			
Male	9 (41%)		
**Ethnicity**			
Chinese	16 (73%)		
Indian	1 (5%)		
Others	5 (22%)		
**Education**			
Tertiary	20 (91%)		
Secondary	2 (9%)		
**Profile of Nature Exposure** [41]			
High (92%–100%)	0		
Versatile (70%–91%)	3 (14%)		
Unilateral (32%–69%)	19 (86%)		
Average (19%–31%)	0		
Low (0%–19%)	0		
**Beck Depression Inventory-II**			
Low (1–16 pt.)	19 (86%)		
Moderate (17–30 pt.)	3 (14%)		
Significant (31–>40 pt.)	0		
**Inter-session break**	M = 11 days (SD = 13 days)		
	Site 1	Site 2	Site 3
Temperature (°C)	M = 29.38 (SD = 0.63)	M = 27.73 (SD = 0.50)	M = 28.09 (SD = 0.62)
Humidity (%Rh)	M = 69.42 (SD = 0.62)	M = 72.13 (SD = 1.37)	M = 68.01 (SD = 0.30)
TDI	M = 26.76 (SD = 0.49)	M = 25.79 (SD = 0.41)	M = 26.52 (SD = 0.54)

Notes: Interpretation of TDI values—<21—no discomfort; 21–24—less than half population feels discomfort; 25–27—more than half population feels discomfort; 28–29—most population feels discomfort and deterioration of psychophysical conditions; 30–32—the whole population feels a heavy discomfort; >32—sanitary emergency due to the very strong discomfort, which may cause heatstroke.

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
