# Peer review of "Can Exposure to Certain Urban Green Spaces Trigger Frontal Alpha Asymmetry in the Brain?—Preliminary Findings from a Passive Task EEG Study"

_ijerph, 2020, doi:10.3390/ijerph17020394_

Round 1
Reviewer 1 Report
An interesting paper aiming to address nature (or green space) exposure and EEG neuro-imaging, in the context of high density Singapore.
The authors applied reliable procedures of collecting neuro-images and causal analysis (more samples would be better), to provide understanding of the seldom unfolded evidence, in term of mental health and green space. It is expected to explain the mechanism of citizens’ outdoor wellbeing, bolster the greening policy in Singapore. The following may be necessary to improve the manuscript.
1. More explanation of mean values of FAA across participants with three sites, to further demonstrate the significance. For example, how your results compare to other research if there were?
2. What particular new theoretical and methodological contributions the study makes need to be more clearly spelled out in the sections of introduction and discussion. The logic need to more tight. The following literature may be relevant, but not necessarily to cite.
Donovan GH, et al. Association between exposure to the natural environment, rurality, and attention-deficit hyperactivity disorder in children in New Zealand: a linkage study. Lancet Planet Health. 2019; 3: e226-e234
Shanahan, D., Bush, R., Gaston, K. et al. Health Benefits from Nature Experiences Depend on Dose. Sci Rep 6, 28551 (2016) doi:10.1038/srep28551
Matilda van den Bosch . Live long in nature and long live nature! Lancet Planetary Health . Volume 1, ISSUE 7, Pe265-e266, October 01, 2017
Reviewer 2 Report
This work mainly evaluated the applicability of using EEG frontal alpha asymmetry (FFA) to objectively correlate with affective feeling of urban green space exposures. 22 individuals were recruited to participate in the landscape scene-viewing experiment while their EEG data were recorded. The results reported some tendency between FFA and landscape score (CLS) towards the posed hypothesis. This issue is interesting and may lead to a new real-life EEG-based brain-computer interface application. However, some major issues required further clarification.
Using EEG signals to characterize affective responses may encounter salient intra- and inter-individual differences, especially for a natural task like the landscape viewing. In addition to exploring averaged FFA of distinct sites, it may be informative to apply regression analysis to exploit the tendency of FFA value per 1-min trial versus the corresponding CLS value.
The experimental procedure asked the subject to complete the self-reported mood states questionnaire before and after each session. This work has to further clarify how the FFA values correlated with those POMS values and how these relationships relate to the CLS values. Otherwise, the current results are plausibly intertwined by ineffective scene trials that were contributed by certain individuals.
The landscape viewing is not likely eliciting time-locked and/or phase-locked EEG oscillations. Thus, it is problematic to average two 1-min long EEG epochs recorded in the same scene before calculating the FAA values.
The more detailed EEG or FFA oscillations are required to report, for example, the EEG changes over 1-min scene viewing versus different conditions (affective state or sites). This may also help to answer if different time windows lead to distinctive outcomes.
This work has to report the weather variables (temperature and humidity) for each session. These factors may affect scene-viewing quality.
Reviewer 3 Report
A good presentation of preliminary data for an important, uncommon metric (FAA) in this field of research. Well written, simple design. Findings are non-significant with a low powered (n=22) preliminary study. Authors may want to retract their submission and resubmitted when the completed study (n=100) is more highly powered and may provide more statistically significant results.
Only a few minor comments about the writing:
1) The abstract provides more information than necessary, consider summarizing some of the abstract and leave the details for the paper.
2) Explanation of the results can be enhanced, particularly regarding Figure 3b. Many readers will not be famliar with FAA and could benefit from an explanation.
(See pdf for comments)
